# Occupational Pesticide Exposure Risks and Gendered Experiences Among Women in Horticultural Farms in Northern Tanzania

**DOI:** 10.3390/ijerph22101529

**Published:** 2025-10-06

**Authors:** Baldwina Olirk, Simon Mamuya, Idda Mosha, Bente Elisabeth Moen, Aiwerasia Ngowi

**Affiliations:** 1Department of Occupational Health and Safety, Muhimbili National Hospital, Dar es Salaam P.O. Box 65000, Tanzania; obabrab@gmail.com; 2Department of Environmental and Occupational Health, School of Public Health and Social Sciences, Muhimbili University of Health and Allied Sciences, Dar es Salaam P.O. Box 65001, Tanzania; mamuyasimon2@gmail.com (S.M.); vera.ngowi@gmail.com (A.N.); 3Department of Behavioural Sciences, School of Public Health and Social Sciences, Muhimbili University of Health and Allied Sciences, Dar es Salaam P.O. Box 65001, Tanzania; ihmosha@yahoo.co.uk; 4Department of Global Public Health and Primary Care, University of Bergen, 5020 Bergen, Norway

**Keywords:** gender roles, horticulture, pesticide exposure, Tanzania, women

## Abstract

Over the past decades, women’s participation in horticulture has become increasingly apparent across Africa. Women perform physically demanding agricultural work on family farms, as hired laborers, or as paid workers on other farms. To increase yield and protect crops, pesticides are used, yet the health risks faced by these women remain under-researched. This qualitative exploratory case study conducted in 2023, in four villages in northern Tanzania, explored pesticide exposure risks, gender roles, and awareness among women working on horticultural farms. Data were collected through four focus group discussions with 46 women (mean age, 39 years; mean work experience, 10 years). Data was transcribed and thematically analyzed. Six themes emerged: gender division of labor, limited training and awareness, adverse health effects, unsafe storage and disposal practices, inadequate protective measures, and resilience. Although pesticide spraying was typically performed by men, poor or unmarried women also undertook this task. Women had limited access to training on safe pesticide handling, and protective gear was rarely used. Despite awareness of potential health risks, economic necessity and prevailing gender norms compelled continued exposure. Gender-sensitive interventions including targeted occupational health education and promotion of safer agricultural practices are urgently needed to reduce pesticide-related health effects among women.

## 1. Introduction

Over the past few decades, women’s engagement in agriculture, including horticulture, has been increasingly intense and apparent in most of Africa [1]. As a result of the commercialization of agriculture here, women make up a significant component of the workforce, particularly in the horticultural sector [2]. The workforce is estimated to be 50–60% by the Food and Agriculture Organization [3] and 47% as per the International Labour Organization [1]. In many African nations, women perform a significant amount of horticultural labour, including 60% of harvesting and marketing and 90% of weeding and land preparation [4].

To intensify horticultural production, farmers use large quantities of pesticides [4]. Most harmful and persistent pesticides are used in horticulture as a result of market liberalization, the privatization of horticultural inputs, and the ineffective enforcement of pesticide laws in most of the African continent [5]. These pesticides, which are fairly priced and widely utilized in African horticulture, are nevertheless illegally accessible through parallel open markets [6]. Similar patterns of dishonest pesticide practices have been documented in several African countries. For example, a study in South Africa revealed that 92% of women were involved in street vending of pesticides without any training [7,8]. In Ethiopia, 93% of women farmers worked on pesticide-sprayed farms without any personal protective equipment (PPE), and none had received formal training on safe pesticide handling [2]. A study in Ghana reported similar findings [9]. In Tanzania, studies indicate that 30–50% of women engage in unsafe farming practices, such as entering recently sprayed fields and washing pesticide-contaminated clothes or equipment without PPE [10]. These practices increase their risk of adverse health effects, including respiratory infections [3] and adverse fetal outcomes [4].

These interactions clarify that women are especially at risk in home or horticultural environments. Due to their lack of education [5,8], knowledge or experience [9], and safety gear [2,10], which is made worse by poverty, they take on risky responsibilities. The women frequently work unprotected in settings that use potentially dangerous pesticides. Furthermore, for biological reasons, women are more vulnerable to pesticides than men. They are more vulnerable to pesticide dangers during pregnancy, lactation, or menopause because of hormonal changes or relatively larger levels of adipose tissue, which can result in acute-to-chronic disorders [4]. During pregnancy or lactation, children of exposed mothers may potentially suffer from developmental problems and other severe illnesses [11]. Even though African women contribute significantly to horticulture, pesticide use, exposure, and the negative health effects they experience have not received the attention they merit [8].

Male farmers have been the subject of the majority of occupational research in agriculture [8,12], and a large portion of the data comes from studies conducted in developed nations with strict regulations on the use of pesticides [12]. Several researchers use the gender division of labor in African agriculture as justification, frequently considering only men and male pesticide applicators to be occupationally exposed [13,14]. Also, it is frequently considered that women are less exposed to pesticides, and this is greatly understated [4]. For many rural African women, this is a misconception. However, despite the widespread belief that men perform the majority of pesticide spraying, research conducted in South Africa [15], Ghana and Mali [9], Uganda [5], and Ghana [16] shows that women play a significant role in the traditionally male-dominated field of pesticide application. Even when women are not directly involved in applying pesticides, they may still be disproportionately exposed through other farm-related tasks. These tasks include working in horticulture farms [17,18] or weeding, harvesting, planting, or packaging in their own fields [5,9]. In addition to their direct exposure to pesticides through agricultural tasks, women may also experience indirect exposure due to their gender-assigned household roles [4]. These include cleaning contaminated clothes and equipment, storing pesticides [5], disposing of empty pesticide containers [8], controlling domestic pests [8], controlling malaria [19,20], or spray drift [21].

Despite this knowledge, policy, research, and advisory services have not adequately addressed the extent and specific nature of pesticide exposure and its associated health effects for women working in horticultural settings in sub-Saharan Africa, including Tanzania [22]. Existing policies and regulations have predominantly focused on men, often assuming that women have minimal exposure [4]. However, this overlooks women’s direct and indirect contact with pesticides through both field and domestic roles [19]. There remains a significant gap in evidence regarding how gendered labor patterns, socioeconomic vulnerabilities, and limited institutional support contribute to exposure and health risks among horticultural women [22].

Our qualitative investigation builds upon earlier quantitative research in the same area, where we found a strong correlation between prenatal pesticide exposure and low birth weight in children born to horticultural mothers [22]. These findings highlight the urgent need to understand the lived experiences, behaviors, and constraints faced by these women. In addition, a previous study emphasized the need for training as a strategy to improve pesticide safety among women working in small-scale horticulture [23]. Despite this previous recent paper, an investigation in a large-scale horticultural setting, where differences in crop production intensity and pesticide use practices are expected, is essential. As such, critical aspects such as the detailed exploration of pesticide storage and disposal practices, the resilience of pregnant women continuing to work despite health risks, and the financial and social pressures shaping women’s behavior were not adequately captured in the previous study. Furthermore, this study was warranted because the experiences of women horticultural workers exposed to pesticides in Tanzania have raised significant public health concerns. This study addressed this gap by exploring risks of pesticide exposure and gender roles in horticulture, as well as awareness of these risks among women working on horticultural farms in Northern Tanzania.

## 2. Materials and Methods

### 2.1. Theoretical Framework

This investigation was guided by the Gender and Development (GAD) Theory, which emphasizes how gender roles are socially constructed and the power disparities that result from them [24]. GAD theory shifts from considering women as a homogeneous, weak group to focusing on how structural elements like decision-making power, labor divisions, and resource availability influence women's experiences in society. GAD provides a lens through which to view how women are exposed to pesticide risks in horticulture due to gendered labor roles, biological vulnerability, frequently without the training, protective gear, or decision-making power to mitigate those risks (Figure 1). The theory emphasizes how wider socioeconomic and cultural limitations influence women's awareness, behavior, and reactions to pesticide exposure rather than being purely a matter of personal preference.

The framework shows how gendered social and economic structures and biological vulnerabilities interact to affect women’s occupational health outcomes in pesticide exposed farms. This vulnerability is largely caused by social and economic factors, including low pay, a lack of other work options, restricted training opportunities, and gender norms that normalize hazardous working conditions for women. Due to their lack of legal protections and frequent exclusion from decision-making, women in rural horticultural settings are less able to negotiate for safer working conditions and are further dependent on their employers. Significant risks of adverse effects from pesticides on occupational and reproductive health, such as signs of pesticide toxicity, low birth weights, and long-term chronic health problems, may result from the interaction of these social and biological forces.

### 2.2. Study Design and Settings

We conducted a qualitative exploratory case study design among women working in horticultural farms between 6 March and 7 April 2023. The case study design was carefully chosen to reflect real-world scenarios [25] where women participate in exhaustive manual labor, usually for longer periods, and are routinely exposed to pesticides in farm activities including pesticide spraying, pesticide mixing, washing pesticide contaminated equipment, weeding, harvesting, storage and disposal processes frequently without adequate protective equipment or training. Using a qualitative strategy might lead to a deeper understanding of gender roles in horticulture, awareness and pesticide exposure risks among women working in horticultural farms in Northern Tanzania. This study was conducted in four villages (in Karatu district: Laghangareri, Barazani, Qang’dend and Mbuganyekundu of Mang’ola and Baray Wards) in northern Tanzania (Figure 2). The study site was selected because it has a considerable number of women employed in agriculture, particularly in horticulture, where women account for 65% to 70% of labor force, and where intensive farming is practiced, and pesticide use occurs year-round [10].

### 2.3. Recruitment of Study Participants and Sample Size

Participants were selected through purposive sampling. We actively involved the local community by consulting village extension officers from the village executive offices, who assisted in identifying and facilitating contact with women actively engaged in horticultural tasks for the focus group discussions (FGDs). We began by identifying the four farms, one from each of the four villages. Women were approached either from homes, farms or during community gatherings. This method was selected to ensure the inclusion of women with direct experience in horticulture and pesticide use. This sampling method was essential to identify information-rich individuals who could meaningfully contribute to the exploration of the study objectives. Therefore, women who are involved in horticultural farming and working in horticultural farms, above 18 years old and have resided in the area for a minimum of one year were recruited by the Principal Investigator (PI). The one-year residency criterion was set to ensure that participants had substantial experience with farm activities and pesticide exposure in the area, rather than only brief or occasional involvement. The recruitment was performed with the help of the community/local leaders and agricultural officers who were familiar with the women with more experience in horticulture farming. All eligible participants approached agreed to participate, and no follow-up interviews were undertaken. In total 46 study participants were recruited.

### 2.4. Data Collection Tools

A total of four FGDs were organized, one from each village, and each consisting of 8–12 women. A pre-tested focus group guide with semi-structured questions was used to collect data. A pilot study was performed in collaboration with the local community involvingone FGD including six participants in a community-level small-scale horticultural farm in Karatu District. Community members helped facilitate participation and provided inputs such as improving clarity and cultural appropriateness. Adjustments were made including corrections of the language for clarity, flow or sequence of the questions and timing. The participants in the pilot study were not included in the main study. The questions in the guide focused on gender roles in horticulture, awareness and pesticide exposure risks among women working in horticultural farms in Northern Tanzania (Table 1).

The focus group guide was written in English and translated into Kiswahili by the PI. The focus group discussions were held in Kiswahili, Tanzania’s national language, for universal accessibility. The PI moderated the FGDs with assistance from a research assistant who took notes of the discussions, organized the use of a tape recorder and kept the time. Each day, a single FGD was held. All FGDs were tape-recorded after the participants had provided their consent. Each FGD lasted approximately 40–60 min, until saturation of the answers from the participants was obtained.

### 2.5. Data Analysis

The recorded discussions were transcribed verbatim. NVivo 14 (QSR International Pty Ltd., Cardigan, UK) was used to organize transcripts, coding text segments, and manage emerging themes across focus group discussions. The initial step involved identifying codes, which were then grouped into categories based on similarities. These categories were further refined to develop overarching themes that conveyed key messages from the data. Two separate experts with expertise in qualitative studies coded the data in addition to the researchers, and their codes matched the themes the researchers had developed. When multiple encoders conduct an encoding, inter/coder dependability must be calculated. Thus, to assess inter/coder agreement, Cohen’s Kappa coefficient was computed. According to the literature, “poor fit” is defined as having a fit value of 0.20 or less, “below-average fit” as having a fit value between 0.21 and 0.40, “moderate fit” as having a fit value between 0.41 and 0.60, “good fit” as having a fit value between 0.61 and 0.80, and “very good fit” as having a fit value between 0.81 and 1.00 [26]. In this investigation, the fit ratio was determined to be 0.9. This number showed that there was excellent agreement across the researchers. Personal relationships had no bearing on the data collection procedures because there was no subjective relationship between the participants and the researchers.

### 2.6. Trustworthiness of the Findings

To guarantee the study’s findings’ reliability, the participants’ descriptions were taken into account when the data were analyzed. Credibility made sure that the results section included the thoughts and viewpoints of every participant. After the discussions, an impartial translator translated the concepts and citations from Kiswahili into English to increase credibility. The researchers compared, examined, and made changes to the translated version. The English version was then translated back into Kiswahili by a multilingual individual who was not a participant in the study. After the initial coding and theme. development in Kiswahili, the data were translated into English. To ensure that the translated theme names accurately captured the original meaning, the research team conducted a back-translation into Kiswahili. This helped refine the English versions to remain semantically close to the original Kiswahili expressions. Therefore, the possibility of a semantic shift in thematic naming was examined. The researcher was encouraged to apply reflexivity in order to increase transparency and “step back from taken-for-granted assumptions” for rigor [27,28]. The study design, data collection, and analysis were guided by the COREQ checklist to ensure comprehensive reporting and methodological rigor [29].

## 3. Results

### 3.1. Demographic Characteristics of the Participants

A total of 46 participants participated in the focus group discussions; The mean age was 39 (±12.8) years. The mean work experience was 10 years. In all groups, most of the women were married (70%) and had only primary school education (80%) while 20% had secondary education (Table 2).

Eleven key words were developed from the four FGDs (Table 3). Based on the analysis, six themes emerged (a gendered division of labor in pesticide use, training and awareness of pesticide use, health effects of pesticides, pesticide storage and disposal practices, protective equipment and resilience among pregnant women) (Table 4 and Figure 3).

### 3.2. Theme 1: Gendered Division of Labor in Pesticide Use

The study revealed a gendered division of labor in pesticide application. Men predominantly carry out tasks requiring physical strength, such as spraying pesticides with heavy equipment. However, unmarried women and those with limited resources take on these roles themselves where they are paid less compared to men, highlighting the intersection of gender and economic status.

(Participant, 42 years)“*For our village, even women spray pesticides. If she’s single and lacks money to hire people to help her, then she must carry the pump herself*.”

### 3.3. Theme 2: Training and Awareness of Pesticide Use

Farmers’ knowledge of pesticide use varies significantly. A few participants reported receiving formal training facilitated by agricultural officers and non-governmental organizations (NGOs) like World Vision. This training provided guidance on safe storage, proper use, and disposal of pesticides. However, many participants emphasized a lack of formal education, relying instead on reading instructions on pesticide containers. Although some of the participants reported receiving help in understanding pesticide label instructions from agricultural extension officers, the majority could not obtain such services more often.

(Participant, 47 years)“*There is no class training, but you may get knowledge when you buy pesticides and read instructions on the bottle*.”

Another participant added,

(Participant, 31 years)“*We didn’t get training, but some of us try our best to follow the instructions provided on bottles.*” Similarly, one participant added “*No! We haven’t received any training, but we use pesticides simply to protect our crops.*”

This self-reliance reflects the community’s resourcefulness in the face of institutional gaps.

### 3.4. Theme 3: Health Effects of Pesticides

Some participants articulated a collective awareness of the health risks associated with pesticide exposure, ranging from skin irritation, respiratory issues, and severe allergic reactions to more critical effects like cancer and reproductive health problems. One farmer remarked:

(Participant, 28 years)“*Some people get blood cancer or throat cancer... Pregnant women have delivered dead babies*.”

It was mentioned by one participant that:

(Participant, 53 years)“*We normally experience effects on our bodies, such as itching, abdominal discomfort, chest tightness, persistent coughing, and severe flu that doesn’t heal easily*.”

### 3.5. Theme 4: Pesticide Storage and Disposal Practices

The majority of participants shared similar perspectives on pesticide storage practices. Some farmers reported hanging pesticides on the roof, in the attic, while others buried them in the farm until the next spraying cycle. Those with designated storage room reported to use them when available, but due to space limitations, some farmers stored pesticides in rooms where they slept. Many participants, however, reported purchasing pesticides and taking them directly to the farm for immediate use.

(Participant, 48 years)“*Many people buy pesticides from the shop and take them straight to the farm. In our village, we put pesticides in a small bag and hang them on the roof where children cannot reach. Honestly, everyone has their way of storing pesticides, whether in a storage room or elsewhere*.”

Some participants emphasized the lack of designated disposal sites for pesticide waste, forcing individuals to store or dispose of by-products in ways that suit their environment:

(Participant, 35 years)“*You have to keep your bottles somewhere and decide what to do with them. If you choose to burn them, that’s up to you. There’s no designated dumping site for disposal, so everyone keeps them based on their convenience*.”

### 3.6. Theme 5: Personal Protective Equipment

Farmers acknowledged the importance of personal protective equipment (PPE), such as gloves, masks, and boots, but cited financial barriers and limited availability as significant challenges. Most rely on improvied methods or forego protection altogether. A participant stated:

(Participant, 51 years)“*We rarely use protection because of lack of money to buy equipment*.”

### 3.7. Theme 6: Resilience Among Pregnant Women

Pregnant women continue to perform all farm activities until they are close to giving birth or feel too fatigued to continue, typically around the seventh month of pregnancy. It was noted that pregnant women continued to work until term and sometimes would give birth while on the farm.

(Participant, 29 years)“*A pregnant woman must work to buy food and children’s needs. She works until she cannot anymore, sometimes until delivery*.” Some of the participants mentioned that some pregnant women will continue to perform light activities like cutting onions “*For tasks that are not too difficult for us, like cutting onions, we sit down, stretch legs, and continue to portion the onions*.”

Notably, many participants repeatedly mentioned that pregnant women remained active in farm work during pregnancy.

(Participant, 47 years)“*In our area, when a woman is pregnant, she continues to work until she gives birth. There is no option to stop any job; she carries out tasks like cleaning or removing weeds, regardless of whether the crops have been treated with pesticides*.”

One participant added,

(Participant, 45 years)“*One day, as I was passing by, I saw a pregnant woman working in the fields. When I returned, I found her surrounded by other women, in labour. They quickly rushed her to the hospital*.”

The majority of study participants reported that after childbirth, women typically resume farming within a few weeks or months, underscoring not only the centrality of horticulture to their livelihoods, but also the financial constraints they face and the lack of systemic support for maternal care.

(Participant, 40 years)“*Those of us who don’t have a partner or are unmarried have no one to support us during pregnancy or after childbirth. So, just a few weeks after giving birth, we must wake up early around 6 a.m.to go to the farm and work to earn money for our daily needs. In the evening, we may be paid two or three thousand shillings, which we use to buy necessities for our children*.”

In addition, another respondent argued,

(Participant, 43 years)“*Even after giving birth, the break is usually just two months. It depends on whether you have food to provide for yourself. What else can you do? You have to get back to work. For me, I only took two weeks off and was back in the fields by the third week*.”

## 4. Discussion

The findings show that economic constraints, a shortage of protective equipment, and structural injustices are the contributing drivers to pesticide exposure among women horticultural workers. The expectation that women, including pregnant women, engage in agricultural tasks like pesticide spraying was a clear example of the gendered division of labour.

Although certain jobs, like spraying chemicals, were historically performed by men, economic necessity has forced many women to take on these responsibilities when assistance is scarce. This finding is in line with those from an investigation in Washington State which reports that pesticide exposure was higher in areas with higher proportion of individuals living below poverty line [28]. In this investigation, it was revealed that individuals with poverty levels exceeding 35% showed an increasing tendency for pesticide exposure. Our findings are also consistent with the other previous research that found that low-income individuals are more likely to be exposed to pesticides than the general population [29]. Our results are supported by the past research that shows how economic marginalization frequently pushes women into hazardous jobs, even when they are pregnant [1]. Poverty is one of the significant factors linked to pesticide exposure as pesticide pollution frequently affects rural communities who experience financial constraints [30,31].

The participants’ overall lack of formal training and knowledge regarding pesticide use was a startling finding. The majority of women depended on informal knowledge or instructions found on pesticide containers, which is consistent with earlier studies that demonstrate that safe pesticide handling training is frequently insufficient or nonexistent in low-income agricultural communities [28,29]. However, relying on the label alone does not guarantee comprehension, especially considering varying education and illiteracy levels [6]. Furthermore, a previous study revealed that agricultural workers in most occasions read the container labels only to identify product names or expiry dates, while they remained unfamiliar with hazard symbols and precautionary statements [32]. In addition to raising the risk of inappropriate handling and exposure, the knowledge gap also reveals a systemic disregard for women’s occupational health education needs [4]. Following pesticide exposure, participants frequently reported effects like skin irritation, respiratory issues, and severe allergic reactions to more critical effects like cancer and reproductive health problems. These results align with previous research showing that pregnant women working in agricultural settings may be exposed to pesticides, which can have both short-term and long-term negative effects on their health, including adverse reproductive outcomes, particularly spontaneous abortion, preterm birth, and stillbirth [29,30,31].

Improper storage and disposal processes, particularly in residences with small living spaces and poor waste management systems, exacerbate the risks of pesticide exposure. Women admitted that used pesticide containers were routinely disposed in hazardous ways or left out in the open, and they described different makeshift storage methods. Similar rural settings lacking regulatory oversight and rising environmental contamination concerns have seen similar behaviors documented [33,34]. In addition to horticultural settings, there have been reports of dishonest pesticide practices, such as the street sale of pesticides in South Africa, which contribute to urban youth poisoning [7]. It is particularly alarming when pregnant women fail to wear personal protective equipment (PPE) during contact with pesticides. Cost, discomfort, and ignorance were among the reasons given. This is consistent with research from other Sub-Saharan African settings where sociocultural and economic barriers prevent women from using PPE [33]. Following safety regulations is made more difficult by the lack of gender-appropriate protective gear.

The tenacity of expectant mothers in these environments was among the most prominent themes. Many women continued working in the fields within weeks of giving birth, often bringing their infants along and feeding them without washing their hands, despite the physical strain. Such tenacity is commendable, but it also reveals underlying structural injustices. Important concerns regarding gender equity, occupational health rights, and social protection mechanisms are brought up by the pressure placed on women to maintain household income even during vulnerable reproductive periods [35,36].

These findings highlight how urgently focused interventions are needed. These could include funding community health education initiatives, enforcing workplace protections for expectant employees, and providing gender-sensitive training on safe pesticide use. Additionally, national occupational health policies need to be expanded to cover workers in the informal sector, especially women in agriculture who continue to be largely invisible in policy frameworks [37,38].

### 4.1. Study Implications

The study’s findings have a number of implications for future research, policy, and occupational health. Despite being aware of the risks, women continue to use hazardous pesticides, which emphasizes the urgent need for gender-sensitive interventions. These could include more stringent enforcement of occupational health and safety laws, the availability of reasonably priced and appropriate personal protective equipment, and easily accessible training programs on safe pesticide use. Furthermore, the evidence regarding reproductive risks emphasizes how crucial it is to incorporate reproductive health concerns into workplace safety regulations. These findings can be used by agricultural extension services and policymakers to customize programs that target the social and economic factors that contribute to unsafe pesticide use.

### 4.2. Strengths and Limitations

Our study has a number of strengths. First, this is one of the few qualitative investigations in Tanzania that examines the gendered aspects of pesticide exposure by means of the lived experiences of women who work directly in horticultural farming—a demographic that is frequently underrepresented in occupational health studies. Second, by using FGDs, the study gathers rich, complex viewpoints that improve the validity and comprehensiveness of the results obtained from the previous quantitative investigation in the same area. By using inductive thematic analysis to identify emerging themes, it was possible for insights to emerge naturally from participant voices, guaranteeing that the results are grounded in the actual world rather than being constrained by preconceived notions. Furthermore, a distinct aspect of vulnerability and resilience that is rarely discussed in pesticide-related research is brought to light by the inclusion of majority of women in reproductive age. The data analysis’s credibility is reinforced by the thorough approach to inter-coder reliability, which includes calculating Cohen’s Kappa coefficient. Ultimately, the study provides important evidence to guide context-specific policy and intervention strategies that can promote safer farming methods and more effectively safeguard the rights and health of women in rural farming communities.

Our study used self-reported data, which might be a weakness. However, to obtain information about delicate topics like health risks during pregnancy and pesticide handling procedures, self-reporting is needed. The main weakness for this method might be potential recall bias. Also, the women might hesitate to express all their opinions on sensitive issues. The discussions were therefore performed in closed rooms, with only the researchers and the women present, underlining confidentiality. Our findings offer valuable insights into pregnant women’s lived experiences in Tanzanian horticultural settings. As agricultural and cultural practices may vary, caution should be taken when extrapolating the findings to other areas or nations. However, the information gathered from the study is likely to be relevant also for other African countries.

Future studies are suggested to examine comparable exposures in various rural settings, use longitudinal designs to evaluate health outcomes over time, and look into ways to lower pregnant women’s exposure to pesticides. Intervention studies after new, gender-oriented regulations will be of interest.

## 5. Conclusions

This study shows that women working in horticulture face substantial occupational risks, including lack of training, safety precautions, and limited legislative protections. Despite experiencing symptoms of pesticide toxicity and being aware of these dangers, women often continue working under hazardous conditions, largely driven by low wages and deeply entrenched gender norms.

The findings highlight how socio-economic pressures and gendered labor roles contribute to persistent occupational health vulnerabilities. The fact that many women remain productive at work throughout their reproductive cycle underscores both their resilience and the urgent need for targeted support. Although the current findings are study-context-specific, they may be informative for similar horticultural settings in Tanzania and comparable regions. To address these challenges, comprehensive and gender-sensitive interventions are essential. These should include accessible training programs, enforcement of occupational health and safety regulations, provision of affordable and suitable personal protective equipment, and the active inclusion of women in the development of occupational and reproductive health policies. Future research should investigate the long-term health effects on these women and explore how legislative strategies can reduce pesticide exposure and associated health risks.

## Figures and Tables

**Figure 1 ijerph-22-01529-f001:**
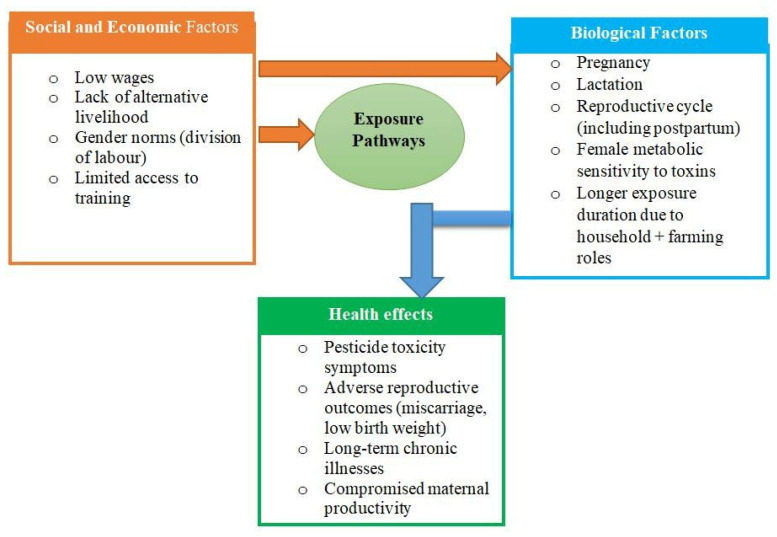
Gender and Development (GAD), a theoretical framework to understand the relation between socioeconomic factors including gender norms, pesticide exposure and health [24].

**Figure 2 ijerph-22-01529-f002:**
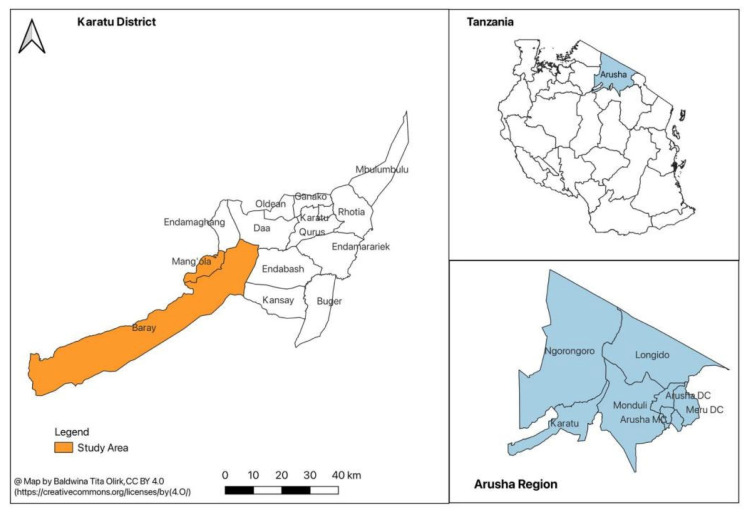
Geographical location of Baray and Mang’ola Wards in Karatu District, Tanzania.

**Figure 3 ijerph-22-01529-f003:**
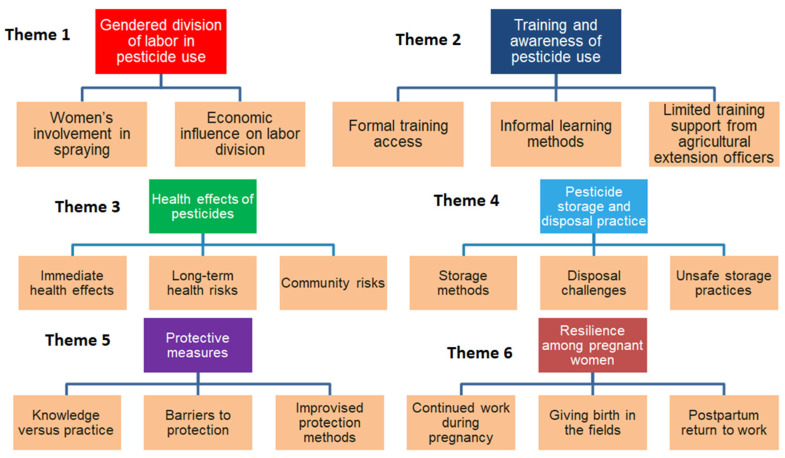
Visual summary of emerging themes and subthemes from focus group discussions among women working in horticulture in northern Tanzania.

**Table 1 ijerph-22-01529-t001:** Focus group guide for women working in horticulture.

Question	
1	What are the main crops mostly grown in your village?
2	In agriculture, what are the different roles of men and women? Who allocated these roles?
3	What are the different roles of men and women regarding the use of pesticides in agriculture?
4	When pesticides are stored at home, where do you usually keep them?
5	Which agriculture roles do women continue doing during pregnancy? Do they have contact with pesticides?
6	At which stage of pregnancy does pregnant women get reduced roles in the farms and why?
7	Based on your experience, how long does it take from delivery to returning to the farms? What are the criteria used?
8	Who have taught you about pesticides and health effects?
9	How do you protect yourself from exposure to pesticides?

**Table 2 ijerph-22-01529-t002:** Demographic data of study participants (N = 46).

Character	Frequency (%)	Mean (SD, Range)
**Age (years)**		39 (12.8, 18–74)
18–29	13 (28.3)	
30–41	14 (30.4)	
>41	19 (41.3)	
Years of work experience		10 (3.8, 1–35)
**Education level**		
Primary school	37 (80)	
Secondary and advanced education	9 (20)	
**Marital Status**		
Single	14 (30)	
Married	32 (70)	

**Table 3 ijerph-22-01529-t003:** Keywords emerging from focus group discussions on women’s pesticide exposure in horticulture.

Keywords	FGD 1	FGD 2	FGD 3	FGD 4
Men spray pesticides				
Washing clothes/equipment				
Weeding				
Harvesting				
Spray if not married			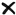	
Carrying pesticide	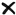			
Domestic spraying				
Digging while pregnant	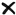	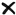		
Harvesting while pregnant				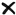
Return soon after birth				
No training				
Learn from others	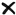			
Extension officer		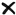	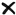	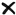
Store pesticides in the kitchen				
Store pesticides under the bed				
Hanging pesticides at the roof top				
Throwing containers in the farms				
Burning pesticides containers				
Gloves	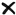			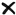
No protection				
Improvized protection (cloth, scarf)				
Low pay/no choice				
No choice/we must eat				
School fees	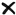	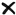		
Headache				
Coughing/Chest pain				
Cancer				
Skin itching/rash				
Children affected				
Dizziness				
Miscarriage				

**Table 4 ijerph-22-01529-t004:** Emerging themes and subthemes from focus group discussions among women in horticulture.

Theme	Subthemes
Gendered division of labor in pesticide use	Women’s involvement in spraying; Economic influence on labor division
Training and awareness of pesticide use	Formal training access; Informal learning methods; Limited training support from agricultural extension officers
Health effects of pesticides	Immediate health effects; Long-term health risks; Community risks
Pesticide storage and disposal practice	Storage methods; Disposal challenges; Unsafe storage practices
Protective measures	Knowledge versus practice; Barriers to protection; Improvied protection methods
Resilience among pregnant women	Continued work during pregnancy; Giving birth in the fields; Postpartum return to work

## Data Availability

The data presented in this study are available on request from the corresponding author due to restrictions from the ethical committees.

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
