# Peer review of "Occupational Pesticide Exposure Risks and Gendered Experiences Among Women in Horticultural Farms in Northern Tanzania"

_ijerph, 2025, doi:10.3390/ijerph22101529_

Round 1
Reviewer 1 Report
Comments and Suggestions for Authors
We believe that the manuscript was written in accordance with scientific principles and the author's field of expertise. We are confident in its novelty and originality. However, several sections of the manuscript require improvement, as noted in the manuscript.

Reviewer 2 Report
Comments and Suggestions for Authors
In the Introduction,
A citation in line 44 should be added. Likewise, a citation is needed in line 52. The sentence from lines 83–91 can also make use of various references. It is recommended that the sources are current and new.
In the methodology section,
It would be helpful to verify if a pilot test of the data collection tool was conducted and detail information on testing validity and reliability procedures. Having visual summaries for all the themes would also enhance clarity and understanding. Including a demographic table indicating participants' age, education, marital status, and work experience in years would also enhance the presentation of results.
Thank you very much.
Comments on the Quality of English LanguageMust be improved.
Reviewer 3 Report
Comments and Suggestions for Authors
Dear Authors, dear Editor,
Draft “2025 ijerph-3867530-pesticides women Tanzania” reports on a survey of information available from interviewing Tanzanian women who cultivate greens and employ pesticides. The topic is itself of the greatest importance in the field of workers and population protection from this occupational hazard.
I find this an excellent draft that nicely shows fundamental topics in the pesticide exposure scenario. A map with the location (Africa – Tanzania – detail of the places) would be a nice addition for foreign readers unaware of geography.
I have only some remarks that derive from my own experience in reporting from focus groups on occupational health topics, which was a peculiar experience for me, since I am a senior chemical laboratory analyst. I only acted as a junior assistant under supervision from an occupational psychologist, so I report from the schedule that I was instructed to follow, which is very similar to that described. the psychologist needed advice on the technical questions, since the topic was about awareness of faculty workers on chemical risk in university research, so the method and aim is fairly close to yours. Therefore, be lenient on my comments.
Do you have a list of keywords that were used to compare answers between the four groups? If yes, the keywords should be listed. We made a table like this
|
Keywords |
Group A |
Group B |
Group C |
|
Keyword |
% of use by participants |
|
|
|
… |
|
|
|
Does it fit?
All the rest of the draft reads very well and comments and conclusions are valuable. I personally agree with the comments.
If you can add details on that, it would be helpful for readers. I mark for minor revisions.
Kind regards
Round 2
Reviewer 2 Report
Comments and Suggestions for Authors
Patient and public involvement should be added.Thank you.
